# Research on the impact of COVID-19 on Chinese small and medium-sized enterprises: Evidence from Beijing

**Zhengwei Ma** *, Yiran Liu, Yida Gao

School of Economics and Management, China University of Petroleum Beijing, Beijing, China

* ma_zhengwei@163.com

## Abstract

COVID-19 leads small and medium-sized enterprises (SMEs) to survive very hard. The development difficulties of SMEs lead to weak employment and GDP growth in various countries. In the process of COVID-19's continuous spread, what is the major reason for the difficulties of SMEs? This paper hopes to answer this question by studying SMEs in Beijing. On this basis, this paper uses structural equation model (SEM) to study the relatively fast recovery of SMEs in Beijing, China, to explore the factors affecting SMEs in the pandemic. After detailed desk research and interviews with relevant entrepreneurs, this paper collects 234 valid questionnaires from SMEs in various industries in Beijing with the help of Federation of Industry and Commerce and Chamber of Commerce in Beijing. Then the data is analyzed with the SEM, which shows the relationship between cash flow from financing activities, markets, employees, costs, government policies and the impact of the pandemic. Finally, an impact model of the pandemic on SMEs is established. The result of the model indicates that the direct effect of the pandemic on the market is the most prominent, and government policies can significantly reduce the negative impact of the pandemic on SMEs indirectly. Based on this, this paper puts forward some policy suggestions, such as the targeted issuance of consumption vouchers and the reduction of administrative barriers. This will enable megacities in various countries to improve policy support for SMEs and promote the recovery and development of SMEs.

## 1. Introduction

### 1.1 Background of the research

The sudden spread of COVID-19 in early 2020 has disseminated rapidly, infecting various industries and sectors in more than 213 countries. Until the end of 2020, the cumulative number of confirmed cases worldwide has reached 91.5 million, and the cumulative death toll has reached 1.96 million. It has seriously affected the lives and health of people around the world and changed the daily routine of the public and society [1]. For instance, loss of small and medium-sized enterprises (SMEs) is particularly severe: According to the report of the China Association of Small and Medium Enterprises, there has been a reduction in business income

**Data Availability Statement:** Data are available at doi.org/10.3886/E149281V1.

**Funding:** The authors received funding from the National Fund of Philosophy and Social Science of China (grant number 18BJY251 to ZM). The

funders had no role in study design, data collection and analysis, decision to publish, or preparation of the manuscript.

**Competing interests:** NO authors have competing interests.

**Abbreviations:** SEM, Structural Equation Model; Factor loading, The correlation coefficient for the variable and factor; Cronbach alpha, A measure of internal consistency, that is, how closely related a set of items are as a group; CR, Construct reliability; AVE, Average variance extracted; GFI, Goodness of fit index; AGFI, Adjusted goodness of fit index; NFI, Normed fit index; CFI, Comparative fit index; RMSEA, Root mean square error of approximation; **Finance**, **Financial condition and cash flow level**; Operating income, The income level of main business operating activities; Sales profit, The profit level of main business operating activities; Solvency capacity, The ability of enterprises to repay external debt; Liquidity stock, The level of emergency liquidity stock; Financing requirements, The amount of external financing requirements of SMEs; **Market**, **Market conditions and basic indicators**; Raw material supply, Supply of raw materials needed for business operations; Market Demand, Demand of consumers for goods and services provided by the enterprise; Product Price, The price at which the company sells goods and provides services; Inventory, The inventory level of the enterprise; Export, The volume of export business and the condition of export default of the enterprise; **Employee**, **Personnel flow condition**; Recruitment, The number of employees recruited by the enterprise; Employee reduction, The number of corporate layoffs; Employee loyalty, The separation rate of enterprise employees; Working hours, Employees' average working hours; Online office, The inconvenience of online office; **Cost**, **Costs incurred due to COVID-19**; Raw material cost, Costs incurred on raw materials; Labor cost, Costs incurred on employees; Online office and transportation costs, Costs of online office and transportation goods; Training cost, The Enterprise's management and training costs; Time cost, Average production cycle of enterprise single product and single service delivery cycle; **Policy**, **Policies proposed by government for COVID-19**; Tax relief, Government tax relief policy; Employment subsidies, Government employment subsidy policy; Operating subsidies, Government operation subsidy policy; Rent Reduction for Commercial Property, Government policy on rent reduction for commercial property; Loan discount, Bank interest-free loan and loan deferred repayment policy; **Impact of COVID-19 on SMEs**| The degree and duration of impact of COVID-19 on

of nearly 67.69% of SMEs; 21.61% of SMEs cannot repay loans and other debts in time, facing greater pressure on operating funds; 86.22% of SMEs cannot survive on funds in their accounts for more than 3 months; 33.73% of SMEs do not have enough funds to survive for one month. The latest data released by the National Bureau of Statistics and the International Monetary Fund (IMF) show that in 2020, the global economy is expected to shrink by 4.4%, the US GDP will drop by 4.3%, the Eurozone GDP will shrink by 8.3%, and Japan's GDP will drop to 5.3%. COVID-19 has a long-term and far-reaching negative impact on the economic development of countries around the world [2]. According to statistics from the International Labor Organization (ILO), the US unemployment rate in October 2020 was 6.9%, with an increase of 3.3% compared to that in October 2019; the EU's overall unemployment rate rose from 6.6% in September 2019 to 7.5% in September 2020; Japan as an advanced economy with the lowest unemployment rate, also witnessed a rise of unemployment rate, from 2.4% in September 2019 to 3.0% in September 2020. The continuing pandemic has worsened the situation with a rising unemployment globally again, leaving the labor market in crisis. In addition, due to the impact of the pandemic, the corporate bankruptcy rate in developed countries is expected to increase by 2.4% in 2020 compared to 1.4% in 2019. All industries in the world have been caught in a state of long-term lockdown and are facing a deadlock of shutdown with no time limit. Therefore, COVID-19 is not only a public health emergency, but also a crisis of global finance and economy. The world will face the double challenges of solving public health challenges and restoring the economy.

## 1.2 Objectives and context of the research

After the outbreak of COVID-19, countries have successively issued corresponding protection policies to control the spread of the pandemic. When the situation just got better, many countries around the world lifted the lockdown in phases starting in early June 2020. New York City began the first phase of "unblocking" on June 8th, with more than 400,000 people back to work, and another five areas in New York State will enter the second phase of resumption of work. The UK started on June 1st, with schools in some areas resuming classes gradually, and "non-essential" retail stores resuming business one after another. In Japan, since August 1st, basic restrictions have been completely lifted, and work has resumed. However, the second wave of pandemic peaked in winter among regions such as the US and Europe. The peak number has exceeded the one in the first wave and is still in the rising stage. This shows that the pandemic has never ended and continuously acted as a hindrance to business management, which has a huge impact towards global public health, economic development, and people's well-being. The Chinese government quickly made strategic deployments in the early stages of the pandemic to provide financial, material and human support for epidemic prevention and control. In this way, the spread of the pandemic in China was controlled to the greatest extent [3]. At present, China has initially achieved the goal of stabilizing the situation and has fully pushed forward the resumption of work and production under normalized prevention and control. As of July 2nd, various industries in China have basically resumed production, with a resumption rate of 99.1%, and on average, 95.4% of people have resumed their work. More than 50 cities have resumed full operations. Among them, the resumption rate of SMEs has reached 91%, which has fully promoted the recovery of production capacity and directly affected economic growth. In 2020, China's GDP is expected to grow at 2.3%, ranking first in the world. This fully reflects the rapid economic recovery of China and its gradual improvement in the prevention and control of pandemic. The experience will help promote international exchanges in medical and healthcare fields, drive the recovery of the world's largest economies, and maintain the stable development of the global economy.

SMEs|Impact of COVID-19 on enterprises themselves, The degree of impact of COVID-19 on SMEs; Persistence of COVID-19's impact, The duration of impact of COVID-19 on SMEs; Impact of COVID-19 on surrounding enterprises, The degree of impact of COVID-19 on surrounding SMEs.

This paper chooses structural equation model (SEM) as the research method to explore the specific impact of the pandemic on Chinese SMEs. The SEM is a technology that tests the causal relationship between variables by combining statistical data and qualitative assumptions. It has been widely applied in the research of various empirical problems as a more comprehensive method of systematic data processing. At the same time, this paper takes Beijing as an example to discover the impact of COVID-19 on Chinese SMEs. With in-depth study of specific influencing factors, and understanding of the impact of the pandemic on all aspects of the production and operation of SMEs, this paper proposes targeted policy recommendations to promote the economic recovery of SMEs, which will do great help to the full recovery of SMEs in megacities in other countries.

## 2. Literature review

### 2.1 Importance of SMEs

With the development of the market economy, SMEs have emerged in large numbers and become the most dynamic group in the market economy. They play an irreplaceable role in profit creation, employment absorption, improvement of people's livelihood, technological innovation and social stability [4]. SMEs refer to organizations that engage in specific economic activities, regardless of the legal form, with certain size restrictions, including individual merchants and family businesses engaged in handicrafts or other activities, as well as partnerships or associations that regularly engage in economic activities [5]. At present, SMEs and self-employed individuals have developed into the main channels for absorbing employment and increasing economic vitality. According to a report of the International Labor Organization (ILO) in August 2019: Statistics from 99 countries show that 70% of employment opportunities are generated by small and micro enterprises and self-employed individuals. The high contribution of small, micro and medium-sized enterprises will help relax social employment pressure, improve people's living standards, and promote harmonious social development. At the same time, the "2020 China's Small and Medium-sized Enterprise Market Status Survey and Development Trend Forecast Analysis Report" released by China Industry Research Network shows that by the end of 2019, Chinese small, micro and medium-sized enterprises accounted for 99.7% of the total number of enterprises in the country. Among them, small and micro enterprises accounted for 97.3%. Chinese SMEs contribute more than 50% of taxation, more than 60% of GDP, more than 70% of technological innovation, more than 80% of urban labor employment, and more than 90% of the number of enterprises [6]. They are the main force of national economic and social development [7]. Since Chinese SMEs are in line with China's resource advantages, they can take a comparative advantage in the competitive market, with strong profitability to create more social wealth. Referring to some data of industrial enterprises from 1993 to 1998, it can be found that SMEs do better than large-scale enterprises in terms of gross production value and the proportion of production value added. This shows that although SMEs have limited scale, they have relatively efficient operating capabilities, which has brought vitality into the market [8].

### 2.2 Influence of COVID-19

**2.2.1 Influence on SMEs in Europe, US and Japan.** At the end of 2019, the sudden spread of COVID-19 disrupted the world economic development and brought huge risks and challenges to the progress of global economics. Multiple pressures such as production stagnation, slow growth in domestic demand, financial constraints, and difficulties in employment caused by the pandemic have severely affected the business activities of enterprises. The challenges are particularly difficult for SMEs with smaller scale and weaker risk tolerance.

According to a survey conducted by the National Federation of Independent Business (NFIB) on March 20th, more than three-quarters of SMEs in the US have been affected by COVID-19. Among them, a survey of 700 companies with 360 employees or less shows that nearly 77% of the currently unaffected companies predict that if the pandemic spreads more widely in the next three months, they would not be spared. Due to the rapid spread of COVID-19, the US has implemented lockdown orders since mid-March, which has brought the economy to a standstill. Within a month, the number of people applying for unemployment benefits for the first time has soared to nearly 22 million, and the unemployment rate of 15% hit the new low since the Great Depression. In addition, the market shrinkage and economic damage caused by the pandemic in Britain, France, Japan and other countries are also extremely serious. Statistics have shown that France's GDP in the first quarter of 2020 fell by 5.3% compared with the previous quarter; Japan's first quarter GDP fell by 3.4% over the previous year; Brazil's first quarter GDP fell by 0.3% year-on-year; UK's first quarter GDP fell to $703.3 billion, decreasing by 1.6% compared with the same period last year. The total production and manufacturing capacity of all countries has declined due to the pandemic. In this context, it is particularly tough for SMEs to make profits. As shown in a McKinsey survey of SMEs in the UK in early May, nearly 80% of the companies surveyed had a decline in their operating income during the pandemic, and the logistics, agriculture and construction industries were most significantly affected by the pandemic. Nearly 77% of SMEs with an annual income of less than 50 million pounds find that the demand for their products and services has fallen, and the supply and demand market has shrunk, which will have a further impact on the national economy. Therefore, the emergence of COVID-19 with high-uncertainty has trapped governments of different countries in the dilemma between saving lives and restoring the economy [9]. SMEs in various countries need to adjust strategies in time to deal with the huge challenges brought by public health issues.

**2.2.2 Influence on SMEs in China.** The spread of COVID-19 has a great impact on China's economy and has brought severe challenges to the survival of domestic SMEs. According to the "Research Report on the Countermeasures and Suggestions on the Impact of COVID-19 Pandemic on SMEs" issued by the China Association of SMEs on February 15th, 2020, nearly 67.69% of SMEs have reduced their operating income; 21.61% of SMEs cannot timely repay debts with increasing pressure on operating funds; 86.22% of SMEs cannot survive with funds on the account for more than 3 months; 33.73% of SMEs do not have enough funds to survive for one month; only 9.89% of SMEs say that they can survive for more than half a year. The fragile financial situation of SMEs backed themselves in a corner, and their trouble directly affected China's economic growth. As a result, China's GDP growth rate in the first quarter fell by 6.8% year-on-year, reaching the lowest in 20 years. At the same time, the job market was sluggish and the unemployment rate rose from 5.3% in January to 6.0% in April [10]. Therefore, the outbreak of the pandemic has continuous influence on Chinese SMEs and the negative impacts stand still. The downward pressure on the economy caused by COVID-19 will not disappear immediately as the pandemic goes to an end. The long-term recovery and the pressure of survival and development remain a task for SMEs.

## 2.3 Influence factors

With the widespread of COVID-19 pandemic, the national economy has been constrained in multiple dimensions. Domestic and foreign scholars have conducted research on the factors affecting the COVID-19 on SMEs. Their perspectives mainly focus on five levels: financing cash flow, market supply and demand, personnel flow, cost, and government policies.

**2.3.1 Financing cash flow.** Through analyzing on the impact of the pandemic on SMEs from a financial perspective, there are four major factors which are corporate income and

profits, solvency, capital liquidity, and financing needs. Severely affected by the pandemic, in the short term, the operating income of SMEs fall sharply, operating costs rise significantly, and operating profits also shrink. According to a survey of more than 5,800 small and micro businesses in New York, after the outbreak of the COVID-19, companies with monthly expenditures of more than $10,000 have cash only enough for two weeks, which largely limits corporations' solvency. In such a period of tight liquidity and cash flow disruption, if without financial support, companies are most likely to face bankruptcy. In the meantime, compared with government grants and corporate loans, ensuring that funds are available to help SMEs recover in a more timely and effective manner [11]. Since SMEs themselves lack access to funds and the pandemic further limits consumer spending, less daily working capital is available to SMEs. Therefore they are less capable to maintain social sustainability, and the goal of keeping market competitiveness becomes even more out of reach [12]. This fully reflects the dependence of SMEs on internally generated liquidity. In the event of a significant public health incident, the disposable liquidity determines the capacity of SMEs to mobilize emergency reserves, that is, the financial capital that enterprises can invest to stimulate production and operation. The more funds invested, the stronger the company's ability to recover, and the smaller impact of the pandemic. However, the new financing needs coming together with the shortage of working capital can help companies repair the capital chain, which fully promotes the resumption of work and production, and restores production capabilities [13]. In time of crisis, financing is crucial to SMEs, and sources of funds are greatly affected by the extreme uncertainty coming together with the pandemic [14]. COVID-19 will have a greater impact on the financial status of SMEs, but establishing a functioning capital structure and ensuring sufficient cash flow will speed up the recovery process of enterprises and promote business development in the crisis.

**2.3.2 Market.**   By the analysis of the impact of the pandemic on SMEs on the market level, it can be concluded that common perspectives mainly include market supply, market demand, commodity prices, inventory levels, exports and defaults. These factors are ultimately based on the adjustment of supply and demand under the market economy system. The pandemic has led to global traffic paralysis because the operations and demand of global airlines have been directly restricted by the government to prevent and control the pandemic [15]. It has severely hindered the transportation of the entire region and further led to insufficient supply of raw materials, a large backlog of inventory orders and other issues. Due to the close relationship among the upstream and downstream of the SMEs' business, upstream suppliers cannot resume production in time, and the resumption of downstream enterprises will also be delayed. This means that the collapse of a single enterprise in the SMEs' supply chain network may lead to the crisis in their upstream and downstream companies at the same time. Most SMEs are private ones and rely mainly on their regular customers. During the pandemic, the government policy restrictions on residents going out and people's worries and panic about getting COVID-19 significantly reduce the consumer demand for non-essential goods, including decreasing expenses on social activities. This leads to less demand for goods and services of SMEs related to the tertiary industry and therefore brings about a difficult time of management [16]. The decrease in supply caused by COVID-19 is mainly due to the uncertain supply of products, inconsistent prices and quality and other issues. The combined effects of unstable transportation channels, slower logistics speeds, drastically reduced export businesses and increased export default orders has brought unprecedented pressure to the survival and development of SMEs [17]. Therefore, because the market did not systematically respond based on different characteristics of corporate governance [18], the pandemic has a relatively large negative impact on the market, which will reduce market supply and demand, leaving enterprises with excessive inventory. At the same time, SMEs have also undertaken more risks of default.

**2.3.3 Personnel flow.** From the perspective of the impact of the pandemic on personnel mobility in SMEs, existing studies are focused on the difficulties in recruiting, large-scale lay-offs, high personnel mobility, reduced working hours, and inconvenience to corporate management caused by online office work. During the pandemic, the loss of a large number of employees has seriously affected the daily operations of the enterprise, and will accelerate the bankruptcy of enterprises to a certain extent. There are more job vacancies in SMEs, especially in high-tech enterprises [19]. In addition, statistical results of relevant data show that remote work is popular during the pandemic. Employees are more adaptable to remote work, but it has brought greater inconvenience to the supervision of managers [11]. At the same time, when the situation relatively gets better, the government has begun to promote enterprises to resume work and production. Since resuming work means an increase in social interactions and an increase in the risk of COVID-19 infection, the perceived work insecurity of employees greatly reduces their work commitment and intention [20]. The pandemic pushes forward the deterioration of the labor market and directly or indirectly restricts the operation of other departments of SMEs. Its impact on the number of employees in SMEs slows down the recovery of the production and processing service chain, and adversely affects the economic recovery of SMEs.

**2.3.4 Cost.** By analyzing impacts of the pandemic on SMEs from the perspective of cost, combined with Brown's application of complex adaptive system theory to develop a comprehensive disaster management framework consisting of economic, social, human, material, natural and cultural capital [21], it can be found that the main focus of this paper includes the cost of raw materials, labor, management training, logistics channel and time. Time cost refers to the cost of extending the average production cycle of a single product and the average service delivery cycle due to the shortage of raw materials and a large number of employees. The extended time also indicates the increasing cost of manpower and resources, reduction of the profitability of the enterprise, and decrease of economic benefits. Although some SMEs in the manufacturing and service industries optimize production and service procedures as much as possible to improve cost efficiency, due to the impact of the pandemic, online management and training of employees are prerequisite, which increases the related expenditures of materials and human capital [22]. In addition, in order to reduce personnel flow during the pandemic, SMEs increase their expenditures on salaries to subsidize employees, enhance their satisfaction and loyalty. At the same time, additional salary subsidies will increase the burden of labor costs. Since fixed costs such as taxes and rents still need to be paid, companies have to bear high operating costs with higher pressure [16]. During the pandemic, poor traffic and port closures increase the cost of logistics such as freight and cargo premium, and the continuity of spare logistics and service networks is greatly influenced [23], which acts as negative impacts towards export trade of SMEs. Large amounts of demurrage and shipment delay fees also need to be paid by the enterprise, which has a significant impact on the operating income of enterprises [17]. The high costs along the supply chain caused by the pandemic have greatly intensified the daily operating pressure of SMEs, which makes it even more difficult to take complex risks without notice, maintain and expand profits, and take advantage of their respective industries to lead the market.

**2.3.5 Policy.** To analyze the impact of the pandemic on SMEs from the perspective of policies, common research mainly focuses on tax and interest reduction, rent relief to commercial properties, and subsidy policies. Under the background of the widespread COVID-19 pandemic, many effective policies are proposed. The British government introduces related policies such as exemption of business rates, suspension of VAT declaration, issuance of interest-free loans, and subsidies for salaries, granting SMEs preferential treatment and protection. The Federal Reserve launched the "Main Street Lending Program" to directly purchase corporate

bonds and lend to SMEs to help companies retain employees and maintain operations [24]. The European Investment Bank will inject 40 billion Euros as liquidity support for SMEs [25]. The part-time job allowance program introduced by Germany aims to subsidize wages of employees through unemployment insurance, thereby reducing frictions when the labor market reopens [26]. Meanwhile, it adds 156 billion Euros of government debt on March 27th, expands the size of the public health budget, and provides payment incentives for small-scale and self-operated enterprises [27]. Based on the impact of preferential policies issued by the Pakistani government on SMEs during the pandemic, it can be found that the government-chartered suspension of commercial property rents reduces the high fixed expenditures of SMEs. At the same time, the government provides subsidies for business-related utilities, such as water, electricity, and natural gas, to release the pressure on business operations and promote the orderly recovery of SMEs [28]. In the context of the pandemic, China is facing tremendous pressure of preventing an economic downturn, which requires proactive fiscal policies to alleviate corporate crises and boost operating efficiency [29]. The government's preferential and subsidy policies for SMEs largely cut down the business spending, reduce the financial pressure on enterprises, and provide significant support for SMEs to survive in the ongoing crisis.

## 2.4 Analytical approach

In this study, the structural equation modeling (SEM) approach was performed to examine the research model. SEM has been known as an advanced estimation technique in business research studies [30].

Hock-Doepgen [31] used SEM to measure the impact of knowledge management capabilities and organizational risk-taking on business model innovation in SMEs; Latifi [32] applied SEM to examine how a firm's performance was affected by innovating its business model; Iwamoto H [33] tested the relationships between human capital, quality management and corporate social performance by using SEM. SEM has been considered as a useful tool to explain and predict constructs in the complex research model. Furthermore, the application of SEM requires fewer demands in terms of sample size as compared to other techniques [34].

## 3. Hypothesis and analysis

### 3.1 Hypothesis and constructs

**3.1.1 Research strategy.** Entrepreneurs have a very real sense of the impact of the epidemic on enterprises. It can be learned from literature review that finance, market, employee, cost, and government policy are crucial factors affecting companies due to COVID-19. Therefore, authors analyzes entrepreneurs' feelings about the impact of COVID-19 on enterprises from these five dimensions. Moreover, structural equation model (SEM) has virtually not been used in existing research to examine entrepreneurs' feelings about the epidemic.

Many researchers select SEM method to study corporate behavior. Malesios [35] used SEM to analyze the sustainability of the supply chain of small and medium-sized enterprises; Viral Gupta [36] used SEM to explore the factors influencing DevOps; Jiwat Ram, David Corkindale and Ming-Lu Wu [37] used SEM to identify the key antecedent factors for accomplishing the adoption stage of enterprise resource planning (ERP) systems. So the SEM is a suitable statistical method to study the state of the enterprise.

The research framework of this study, based on entrepreneurs' perception of the impact of COVID-19 on enterprises, is shown in Fig 1. The theory and research hypotheses of this study are presented with referring to the Theory of Reasoned Action (TRA) [38], the Technology Acceptance Model (TAM) [39] and the Theory of Planned Behavior (TPB) [40].

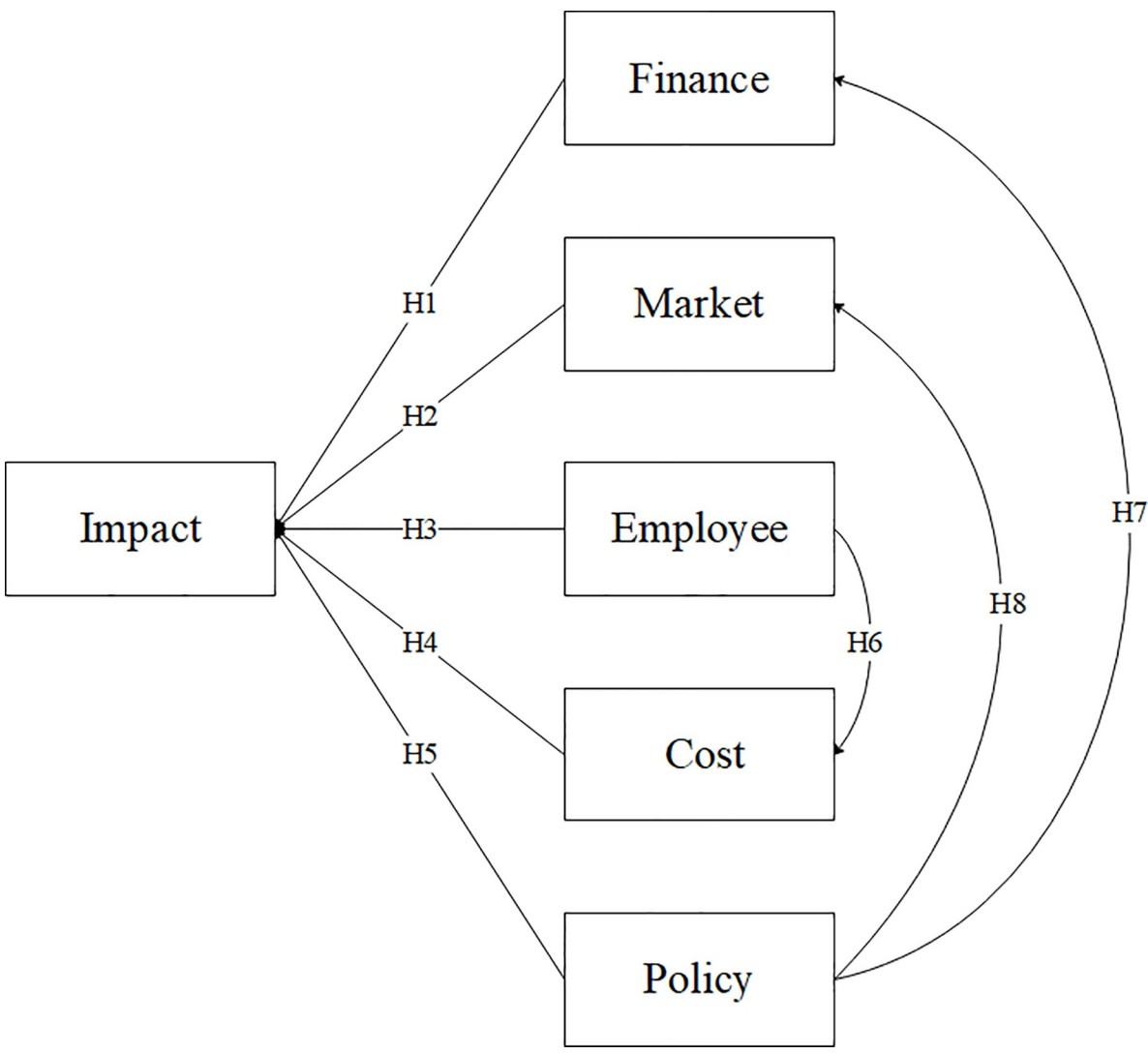

**Fig 1. Research factors and hypotheses in the present study.**

**3.1.2 Hypothesis.** Taking into account the previous considerations, the relationship among finance, market, employee, cost, and government policy is evident in personal data handling and should be examined in greater detail to test the connections with the impact of COVID-19.

Due to the COVID-19 pandemic, the government has issued relevant epidemic prevention and control policies, to restrict the operation of enterprises in many industries. On the occasion that companies need to be closed for a month or more, most SMEs have almost no cash or equivalents to pay rent, salaries, loan interest and other running expenses, which have aggravated the shortage of cash flow and have led to further impacts on the fragile financial situation of SMEs [16]. The capacity of enterprises to withstand fiscal pressures induced by the pandemic explains a widespread loss of staff at the onset of the pandemic [41]. Operational interruptions and increased financing costs brought about by COVID-19 have made it particularly difficult for companies with low risk-resistance in disaster-stricken areas to obtain financing [42]. Therefore, based on the findings of previous studies, this study proposes the following hypothesis:

H1. There will be a positive relationship between problem of finance and negative impact of COVID-19.

COVID-19 spreads around the world, and many countries block their business scope, limit the market size, so that the supply side has been greatly damaged [43]. The biggest and most immediate impact of the lockdown is the halt in business operations. Due to lockdown, supply and demand chain has been disrupted, which causes the reduction in sales and profit [28]. The reduction in demand may also be caused by the decrease of consumers' income, shortage of funds, and severe constraints on their ability to pay [22]. In addition, part of the impact of COVID-19 on the social economy is guided through the labor market, which has a longer-term impact on labor supply [44]. Thus, this study establishes the following hypothesis:

H2. There will be a positive relationship between problem of market and negative impact of COVID-19.

During the pandemic, some SMEs terminated their operations, which has an essential connection with the large reduction in the number of employees caused by the difficulty in recruiting and layoffs. Among them, workers with lower education levels, regardless of their gender, are at higher risk of being laid off, and the decline in their work efficiency due to economic blockade is more obvious [45]. The resumption rate of employees in SMEs is low, working hours and efficiency are significantly reduced, resulting in a shortage of human resources and an increase in labor costs. Therefore, SMEs will face major operational obstacles [16]. This study proposes the following hypotheses:

H3. There will be a positive relationship between problem of employee and negative impact of COVID-19.

H6. There will be a positive relationship between problem of employee and problem of cost.

Based on the experiences, the sudden attack of the pandemic will bring a series of economic shocks. The operating costs, including material, labor and time costs, along the production chain of SMEs will significantly increase in the short term. Many SMEs have signed exclusive agreements with their suppliers, but the shrinking scale of production and insufficient supply of their suppliers during the pandemic force SMEs to ignore price concessions and choose expensive raw materials for business activities, which in turn increases the cost of raw materials [46]. Therefore, this study suggests the following hypothesis:

H4. There will be a positive relationship between problem of cost and negative impact of COVID-19.

After the widespread of COVID-19 pandemic, governments successively introduce policies to restore the normal operation of SMEs, stimulate economic recovery, and promote comprehensive resumption of work and production. When the pandemic continues to spread, the government will reduce bank interest rates and extend the repayment period of enterprises, which reduces the pressure on SMEs' funds and acts as assistance for the restoration of SMEs' liquidity [47]. Reducing the collection of corporate taxes will help cut down operating costs of corporations and stimulate the market to expand. Although it will decline government's short-term budgets and increase the financial pressure of local governments, more tax revenues and employment opportunities shall be created in the long run as companies recover [17]. Financial subsidies, tax relief and other economic assistance have significantly minimized negative impacts of the pandemic, including the shortage of working capital of SMEs and the massive loss of employees. Thus, based on previous research, this study proposes the following hypotheses:

H5. There will be a negative relationship between government policy and negative impact of COVID-19.

H7. There will be a negative relationship between government policy and problem of finance.

H8. There will be a negative relationship between government policy and problem of market.

## 3.2 Materials and methods

### 3.2.1 Data collection

The generation of the initial questionnaire was ascertained by the interviews and in-depth discussions about the impact of COVID-19 with scholars and experts in SMEs. Pre-tests of the initial 28-item questionnaire were carried out with 30 researchers in SMEs to improve the questionnaire. The resulting modified 28-item pool was presented to the questionnaire about the impact of COVID-19. Respondents were asked about their attitudes towards the impact of COVID-19 in the questionnaire. Distributing questionnaires through Beijing Federation of Industry & Commerce, authors exerted non-random method of collecting the data (volunteer sampling) to generate 234 valid questionnaires from experts, scholars or researchers in SMEs, whose opinions could reflect the extent of COVID-19's impact on SMEs in Beijing. The survey results are shown in **Table 1**.

**3.2.2 Exploratory factor analysis.** An exploratory factor analysis using SPSS 25 (SPSS refers to Statistical Product and Service Solutions. It is a software package used for statistical analysis produced by SPSS Inc. 25 means No. 25 version.) was conducted on the first-hand data. The rotated factor matrix, resulting from a Promax rotated principal axis factor extraction of the independent variables using the 1.0 eigenvalue cut-off criterion (see **Table 2**), indicates that twenty-five factors emerged and reports their factor loadings.

The data were tested using the SPSS 25 Exploratory Factor Analysis to evaluate the Cronbach alpha. The Cronbach alpha indicator is the most frequently used test for assessing reliability. The results show the value for finance's Cronbach alpha is 0.946, the value for market's Cronbach alpha is 0.930, the value for employee's Cronbach alpha is 0.936, the value for cost's Cronbach alpha is 0.947 and the value for policy's Cronbach alpha is 0.944. These five values exceed 0.7 and they are of high reliability [48]. This is satisfactory. Each item was evaluated individually to ensure convergent validity and item reliability.

All factor loadings were larger than 0.8, representing an acceptable significant level of internal validity. The factor loadings ranged from 0.902 to 0.915 for finance; from 0.865 to 0.896 for market; from 0.875 to 0.916 for employee; from 0.888 to 0.920 for cost; from 0.897 to 0.919 for policy; from 0.931 to 0.949 for impact of COVID-19. All factor loadings were of an acceptable significant level, thus all twenty-eight items were retained for further analysis (see **Table 2**).

**3.2.3 Confirmatory factor analysis.** Authors developed a structural equation model (SEM) with the objective of testing the proposed hypotheses (**Fig 2**). Authors observed that the hypothesis was supported at the 0.1 level. Model fit was acceptable (Chi–square = 105.471 df = 64, p<0.05, normed Chi-Square = 1.192). Through calculation, authors obtained SEM fit indexes, and listed the processes in the coming paragraphs.

The GFI (goodness of fit index) was devised by Jöreskog and Sörbom [49] for MI and UI estimation, and generalized to other estimation criteria by Tanaka and Huba [50]. The GFI is given by

$$GFI = 1 - \frac{\hat{F}}{\hat{F}_b} \tag{1}$$

**Table 1. The respondents' viewpoints about Impact of COVID-19 on SMEs.**

| | Strongly agree (%) | Agree (%) | Slightly agree (%) | Neutral (%) | Slightly disagree (%) | Disagree (%) | Strongly disagree (%) |
|---|---|---|---|---|---|---|---|
| Operating income | 27.4 | 19.2 | 9 | 12.8 | 13.7 | 9.4 | 8.5 |
| Sales profit | 22.6 | 22.6 | 16.7 | 11.1 | 13.2 | 10.3 | 3.4 |
| Solvency capacity | 25.2 | 18.4 | 14.1 | 15 | 9.8 | 11.1 | 6.4 |
| Liquidity stock | 24.4 | 21.8 | 15 | 10.3 | 15.4 | 8.5 | 4.7 |
| Financing requirements | 20.9 | 20.1 | 15.8 | 12.4 | 14.1 | 9.4 | 7.3 |
| Raw material supply | 18.4 | 20.1 | 18.4 | 14.5 | 11.5 | 12 | 5.1 |
| Market Demand | 16.2 | 22.6 | 17.1 | 15 | 12.8 | 9.8 | 6.4 |
| Product Price | 15.8 | 19.7 | 17.9 | 15.4 | 12.4 | 13.2 | 5.6 |
| Inventory | 18.4 | 17.9 | 15 | 19.2 | 14.5 | 10.3 | 4.7 |
| Export | 17.9 | 20.1 | 19.2 | 15.4 | 10.7 | 9.4 | 7.3 |
| Recruitment | 16.2 | 22.6 | 14.1 | 14.5 | 13.7 | 9 | 9.8 |
| Employee reduction | 14.5 | 20.1 | 16.2 | 16.7 | 14.1 | 12.8 | 5.6 |
| Employee loyalty | 15.4 | 19.2 | 13.2 | 14.1 | 16.7 | 13.2 | 8.1 |
| Working hours | 15.4 | 23.1 | 14.5 | 15 | 15 | 11.5 | 5.6 |
| Online office | 16.7 | 19.2 | 16.2 | 16.2 | 15 | 10.3 | 6.4 |
| Raw material cost | 16.7 | 23.1 | 18.8 | 13.2 | 13.7 | 8.5 | 6 |
| Labor cost | 19.7 | 21.8 | 17.5 | 13.2 | 12.4 | 10.7 | 4.7 |
| Online office and transportation costs | 15.8 | 20.9 | 16.7 | 15 | 14.5 | 9.4 | 7.7 |
| Training cost | 17.1 | 22.6 | 17.9 | 12.4 | 12.4 | 10.7 | 6.8 |
| Time cost | 17.1 | 20.1 | 17.1 | 15.8 | 15.4 | 8.5 | 6 |
| Tax relief | 17.9 | 20.1 | 20.5 | 13.2 | 14.5 | 9 | 4.7 |
| Employment subsidies | 18.8 | 20.9 | 14.1 | 17.9 | 13.2 | 9.4 | 5.6 |
| Operating subsidies | 19.7 | 18.4 | 17.9 | 16.7 | 14.1 | 8.5 | 4.7 |
| Rent Reduction for Commercial Property | 16.2 | 22.6 | 19.7 | 15.8 | 13.7 | 7.7 | 4.3 |
| Loan discount | 17.5 | 20.9 | 17.1 | 17.9 | 13.7 | 5.1 | 7.7 |
| Impact of COVID-19 on enterprises themselves | 20.1 | 20.9 | 14.1 | 16.7 | 13.7 | 8.5 | 6 |
| Persistence of COVID-19's impact | 19.2 | 24.4 | 17.1 | 12 | 12 | 8.1 | 7.3 |
| Impact of COVID-19 on surrounding enterprises | 20.5 | 21.8 | 16.7 | 16.7 | 12.8 | 5.6 | 6 |

Where $\hat{F}$ is the minimum value of the discrepancy function and $\hat{F}_b$ is obtained by evaluating F with $\Sigma^{(g)} = 0$, g = 1,2,...G. An exception has to be made for maximum likelihood estimation, since (D2) is not defined for the purpose of computing GFI in the case of maximum likelihood estimation, $f(\Sigma^{(g)}; S^{(g)})$ is calculated as:

$$f\left(\sum\nolimits^{(g)}; S^{(g)}\right) = \frac{1}{2} \operatorname{tr}\left[K^{(g)^{-1}}\left(S^{(g)} - \sum\nolimits^{(g)}\right)\right]^2 \tag{2}$$

with $K^{(g)} = \sum^{(g)}(\hat{\gamma}_{ML})$, where $\hat{\gamma}_{ML}$ is the maximum likelihood estimate of $\gamma$. By using the formula (1) and (2), authors calculated the Model's GFI as 0.901.

The AGFI (adjusted goodness of fit index) takes into account the degrees of freedom available for testing the model. It is given by

$$AGFI = 1 - (1 - GFI)\frac{d_b}{d} \tag{3}$$

**Table 2. Factor loading.**

| Factors | Factor loading | Cronbach alpha | Construct Reliability (CR) | AVE |
|---|---|---|---|---|
| **Finance** | | 0.946 | 0.9593 | 0.8249 |
| Operating income | 0.907 | | | |
| Sales profit | 0.911 | | | |
| Solvency capacity | 0.902 | | | |
| Liquidity stock | 0.915 | | | |
| Financing requirements | 0.907 | | | |
| **Market** | | 0.930 | 0.9469 | 0.7809 |
| Raw material supply | 0.896 | | | |
| Market Demand | 0.896 | | | |
| Product Price | 0.868 | | | |
| Inventory | 0.893 | | | |
| Export | 0.865 | | | |
| **Employee** | | 0.936 | 0.9514 | 0.7966 |
| Recruitment | 0.875 | | | |
| Employee reduction | 0.916 | | | |
| Employee loyalty | 0.905 | | | |
| Working hours | 0.890 | | | |
| Online office | 0.876 | | | |
| **Cost** | | 0.947 | 0.9596 | 0.8260 |
| Raw material cost | 0.912 | | | |
| Labor cost | 0.913 | | | |
| Online office and transportation costs | 0.920 | | | |
| Training cost | 0.911 | | | |
| Time cost | 0.888 | | | |
| **Policy** | | 0.944 | 0.9572 | 0.8173 |
| Tax relief | 0.897 | | | |
| Employment subsidies | 0.908 | | | |
| Operating subsidies | 0.919 | | | |
| Rent Reduction for Commercial Property | 0.898 | | | |
| Loan discount | 0.898 | | | |
| **Impact of COVID-19 on SMEs** | | 0.935 | 0.9589 | 0.8862 |
| Impact of COVID-19 on enterprises themselves | 0.944 | | | |
| Persistence of COVID-19's impact | 0.931 | | | |
| Impact of COVID-19 on surrounding enterprises | 0.949 | | | |

Where

$$d_b = \sum_{g=1}^{G} p^{*(g)} \qquad (4)$$

Through the use of the formula (3) and (4), authors got that the model's AGFI value is 0.878.

The Bentler-Bonett normed [51] fit index (NFI), or $\Delta_1$ in the notation of Bollen [52] can be written as

$$NFI = \Delta_1 = 1 - \frac{\hat{C}}{\hat{C}_b} = 1 - \frac{\hat{F}}{\hat{F}_b} \qquad (5)$$

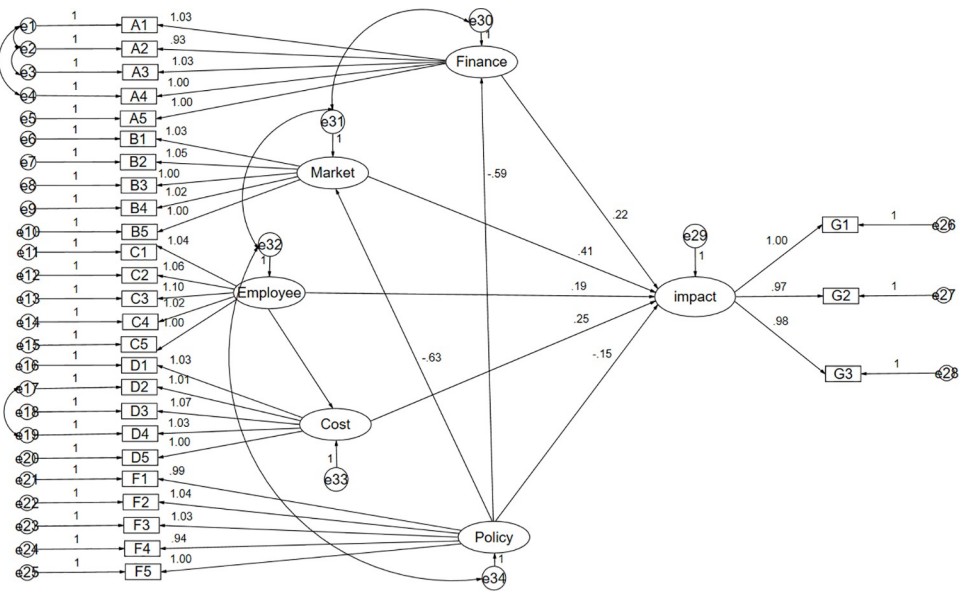

**Fig 2. The structural equation model.**

Where $\hat{C} = n\hat{F}$ is the minimum discrepancy of the model being evaluated and $\hat{C}_b = n\hat{F}_b$ is the minimum discrepancy of the baseline model. By using the formula (5), authors knew that the Model's NFI is 0.943.

The comparative fit index (CFI; [53]) is given by

$$\text{CFI} = 1 - \frac{\max(\hat{C} - d, 0)}{\max(\hat{C}_b - d_b, 0)} = 1 - \frac{\text{NCP}}{\text{NCP}_b} \tag{6}$$

where $\hat{C}$, d and NCP are the discrepancy, the degrees of freedom and the noncentrality parameter estimate for the model being evaluated, and $\hat{C}_b d_b$ and $\text{NCP}_b$ are the discrepancy, the degrees of freedom and the noncentrality parameter estimate for the baseline model. According to formula (6), authors figured out that the Model of the study's CFI is 0.990.

$F_0$ incorporates no penalty for model complexity and tends to favor models with many parameters. In the comparison between the two nested models, $F_0$ will never favor the simpler model. Steiger and Lind [54] suggested compensating for the effect of model complexity by dividing $F_0$ by the number of degrees of freedom for testing the model. Taking the square root of the resulting ratio gives the population "root mean square error of approximation", called RMS by Steiger and Lind [54], and RMSEA by Browne and Cudeck [55].

$$\text{Population RMSEA} = \sqrt{\frac{F_0}{d}} \tag{7}$$

$$\text{Estimated RMSEA} = \sqrt{\frac{\hat{F}_0}{d}} \tag{8}$$

The results show that the RMSEA index is 0.029.

In conclusion, our model exhibited a reasonable fit with the data collected. Authors assessed the model fit using other common fit indices: goodness-of-fit index (GFI), adjusted goodness-

of-fit index (AGFI), normed fit index (NFI), comparative fit index (CFI), root mean square error of approximation (RMSEA) and root mean square residual (RMR). The model exhibited a fit value exceeding or close to the commonly recommended threshold for the respective indices and the commonly suggested value would be listed in **Table 3**.

**3.2.4 Construct reliability analysis.** The construct reliability of the latent variables is an evaluation standard for the inner quality in a structural equation model. If the construct reliability (CR) is higher than 0.7, the inner quality of the model is considered to be acceptable [56]. The authors will use the model standardized regression weights to calculate the construct reliability, presented as $\rho_c$. Construct reliability of finance, market, employee, cost, policy and impact of COVID-19 were calculated at a suggested lower limit of 0.70 with Eq (9). The results have been shown in the **Table 2**.

$$\rho_{c1} = \left[ \frac{\left( \sum \lambda_1 \right)^2}{\left( \sum \lambda_1 \right)^2 + \sum \theta_1} \right] \tag{9}$$

Another index, similar to construct reliability, is "average variance extracted (AVE)," presented as $\rho_v$. This index can explain how much variance explained in the latent variable comes from the observed variables. The higher the average variance extracted, the better the observed variables could explain the latent variable. Generally speaking, the model's inner quality is considered to be good when the average variance extracted is higher than 0.5. The average variance extracted from finance, market, employee, cost, policy and impact of COVID-19 were calculated at a suggested lower limit of 0.50 with Eq (10). The results have been shown in the **Table 2**.

$$\rho_{v1} = \left[ \frac{\left( \sum \lambda_1^2 \right)}{\left( \sum \lambda_1^2 \right) + \sum \theta_1} \right] \tag{10}$$

$$\mathrm{Chi-square} = 439.395$$

$$\mathrm{CMIN/df} = 1.312$$

$$P = .000$$

$$\mathrm{GFI} = .891 \ \mathrm{IFI} = .984 \ \mathrm{TLI} = .982 \ \mathrm{CFI} = .984 \ \mathrm{AGFI} = .868$$

$$\mathrm{RMSEA} = .037 \ \mathrm{NFI} = .936$$

**Table 3. Fit statistics of final model.**

| Fit statistic | Suggested | Obtained |
|---|---|---|
| Chi-square | | 439.395 |
| Df | | 335 |
| Chi-square significance | P<or = 0.05 | 0.000 |
| Chi-square/df | <3 | 1.312 |
| GFI | >0.80 | 0.891 |
| AGFI | >0.80 | 0.868 |
| NFI | >0.90 | 0.936 |
| CFI | >0.90 | 0.984 |
| RMSEA | <0.08 | 0.037 |

**Table 4. Path coefficients and their significance values.**

| Paths | path coefficient | S.E. | C.R. | P |
|---|---|---|---|---|
| Impact of COVID-19 ← Finance (H1) | 0.221 | 0.049 | 4.476 | *** |
| Impact of COVID-19 ← Market (H2) | 0.415 | 0.067 | 6.175 | *** |
| Impact of COVID-19 ← Employee (H3) | 0.187 | 0.065 | 2.871 | 0.004** |
| Impact of COVID-19 ← Cost (H4) | 0.247 | 0.056 | 4.416 | *** |
| Impact of COVID-19 ← Policy (H5) | -0.152 | 0.062 | -2.453 | 0.014* |
| Cost ← Employee (H6) | 0.641 | 0.067 | 9.601 | *** |
| Finance ← Policy (H7) | -0.590 | 0.070 | -8.373 | *** |
| Market ← Policy (H8) | -0.628 | 0.065 | -9.682 | *** |

Note

* $p < 0.05$.

** $p < 0.01$.

*** $p < 0.001$.

## 4. Results

After analyzing the data, the acceptable measurement model has been established. Authors evaluated the structural model shown in a path diagram in **Fig 1** and the path coefficients and their significance values are reported in **Table 4**. Testing the statistical significance of their parameter estimates from SEM, the Critical Value (C.R.), which represents the parameter estimate divided by its standard error (S.E.) is used. Based on a significance level of 0.1, the C.R. needs to be $> \pm 1.65$.

### 4.1. Direct effect

The results of the path analysis show that the direct path coefficient of the impact of financing cash flow on the pandemic is significantly positive ($\rho = 0.221$, C.R. = 4.476, $p < 0.001$), so Hypothesis 1 is valid, which means that the falling income level, the worse ability of enterprises to repay external debt as well as the lack of emergency liquidity stock all contribute to the negative impact of COVID-19 on small and medium enterprises. Business owners' increasing demands for external financing also exacerbate the negative impact of the epidemic. As shown in **Fig 2**, the direct path coefficient of the market's impact on the pandemic is significantly positive ($\rho = 0.415$, C.R. = 6.175, $p < 0.001$), so Hypotheses 2 is valid. It can be seen that during the COVID-19, the supply of materials required for business operation is reduced, consumers' demand for goods and services is decreased, which all hinder the business running of enterprises. In addition, market problems in the epidemic have an impact on the company due to the falling commodity prices and the increasing export business defaults, which means that the impact of market issues has a multi-dimensional impact on the company. As shown in **Fig 2**, the direct path coefficient of the impact of employee turnover on the epidemic is significantly positive ($\rho = 0.187$, C.R. = 2.871, $p = 0.004$), so Hypotheses 3 is valid. It indicates that, during the COVID-19, employee has an impact on the company's own operations. Due to the recession of labor market, enterprises face the problems of recruitment difficulties and high employee turnover rate. At the same time, due to the poor management of the enterprise, a large number of employees suffered layoffs, with a crisis of unemployment. Because of increasing online office work and decreasing working hours, the company's costs have risen during the epidemic. As shown in **Fig 2**, the direct path coefficient of the impact of cost on the epidemic is significantly positive ($\rho = 0.247$, C.R. = 4.416, $p < 0.001$), so Hypothesis 4 is valid. It means that due to COVID-19, the increase in the price of raw materials because of shrinking

market and additional demand for epidemic prevention materials force enterprises to spend more on materials. The staff management problems brought by online office and epidemic prevention work restrictions also cause the company's cost to substantially rise. Furthermore, lack of raw material supply and employee stability, the average production cycle and service supply cycle both have been significantly extended, rising time costs of the enterprises. In addition, the direct path coefficient of the impact of government policies on the pandemic is negative and significant ($\rho$ = -0.152, C.R. = -2.453, p = 0.014), so Hypothesis 5 is also valid (Table 4). The government can reduce the adverse impact of the epidemic on companies through tax incentives, interest-free loans, employment subsidies, operating subsidies, and rent reductions. However, the absolute value of the direct path coefficient of its impact on the epidemic is the smallest compared to finance, market, employee and cost, indicating that the direct impact of the policy on enterprises is limited. (Fig 3 and Table 5).

## 4.2. Indirect effect

Based on research results, the conclusion is drawn that personnel flow and government policies have limited direct impacts on the influence of the pandemic on SMEs, but the low stability of employees and the increasing working allowances and subsidies due to the COVID-19 greatly increase human capital expenditures ($\rho$ = 0.641, C.R. = 9.601, p<0.001), so through increased costs, problems of employees indirectly expand the impact of the pandemic on companies. Government policies have obvious effects on financing cash flow ($\rho$ = -0.590, C.R. = -8.373, p<0.001) and market ($\rho$ = -0.628, C.R. = -9.682, p<0.001). It indicates that the policy of government tax relief and commercial property rent reduction have saved part of the operating expenditures of the enterprise, and the bank interest-free loan as well as the loan delay repayment policy have had a great positive impact on the financial survival and development of small and medium-sized enterprises. Government employment and operation subsidies have also relieved the financial pressure on enterprises to some extent. In addition, the subsidy and welfare policies proposed by the government and banks have played a major role in

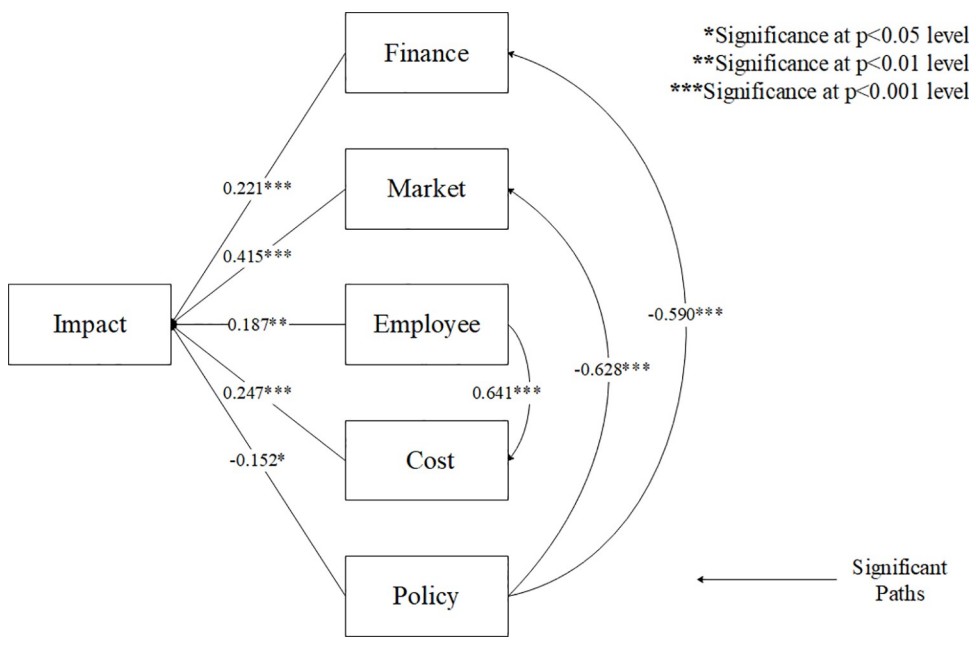

**Fig 3. Path coefficients and effect.**

**Table 5.  Effect on policy, market, finance, employee, cost and impact.**

|  | Policy | Market | Finance | Employee | Cost | Impact |
|---|---|---|---|---|---|---|
| Policy | 0 | TE:-0.641 | TE:-0.387 | TE:-0.256 | Insignificant Paths | TE:-0.735 |
|  |  | DE:-0.641 | DE:-0.387 | DE:-0.256 |  | DE:-0.159 |
|  |  | IE:0 | IE:0 | IE:0 |  | IE:-0.576 |
| Market |  | 0 | TE:0.367 | TE:0.353 | TE:0.144 | TE:0.642 |
|  |  |  | DE:0.367 | DE:0.353 | DE:0.144 | DE:0.415 |
|  |  |  | IE:0 | IE:0 | IE:0 | IE:0.227 |
| Finance |  |  | 0 | Insignificant Paths | Insignificant Paths | TE:0.210 |
|  |  |  |  |  |  | DE:0.210 |
|  |  |  |  |  |  | IE:0 |
| Employee |  |  |  | 0 | TE:0.553 | TE:0.325 |
|  |  |  |  |  | DE:0.553 | DE:0.190 |
|  |  |  |  |  | IE:0 | IE:0.135 |
| Cost |  |  |  |  | 0 | TE:0.245 |
|  |  |  |  |  |  | DE:0.245 |
|  |  |  |  |  |  | IE:0 |

Note: TE = Total effect, DE = Direct effect, IE = Indirect effect.

maintaining the supply chain and stimulating the recovery of market consumption. Therefore, financing cash flow and market both can play intermediary roles. Policies can indirectly but significantly reduce the negative impact of the epidemic on SMEs. Conclusions of the hypothesis test of the indirect influence and the indirect path coefficients are shown in (**Fig 3** and **Table 5**).

To sum up, all the hypotheses proposed are significantly supported (see **Fig 4**).

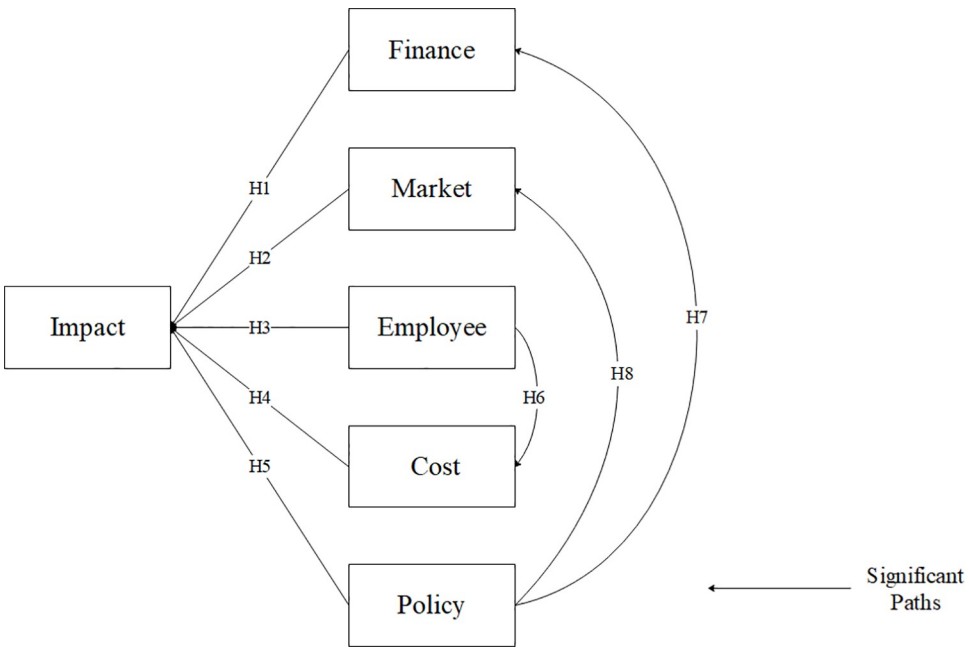

**Fig 4.  Structural equation results model.**

## 5. Conclusion and suggestions

This paper applies the structural equation model (SEM) to the research and finds that financing cash flow, market, personnel flow, cost, and government policies are the major factors that measure the impact of the pandemic on SMEs. Among them, market has the greatest direct impact on SMEs, and government policies provide the greatest indirect assistance to SMEs in the plight of the pandemic. Financing cash flow, personnel mobility, and costs play relatively less important role in exacerbating the negative impacts of the pandemic. Therefore, this paper makes relevant suggestions in terms of promoting market recovery and government policy assistance:

First, issue targeted consumption vouchers, implement targeted consumption preferential policies, and stimulate market consumption expansion. The social isolation caused by the worldwide spread of COVID-19 has triggered a sharp decline in market consumption, which has a greater impact on the internal structure of consumption, and has become major obstacles to the development of SMEs. All localities should actively adopt regular, targeted, and flexible policies and measures for the tertiary industry whose consumption level has dropped significantly to increase residents' willingness to consume and gradually fulfill the consumption potential. Local governments can periodically issue consumption vouchers such as supermarket shopping coupons, holiday commodity coupons, and catering coupons based on latest market and industry news to guide the steady growth of consumption and promote the normalization of social production and life. At the end of 2020, China's retail industry has tentatively recovered from the shock of the pandemic. With the gradual recovery of consumer confidence and the catalysis of various consumption preferential policies, recovery of market demand is in the foreseeable future. In the practice of stimulating market consumption in different countries, the core policy to promote "experiential consumption" is set. The UK government issues offline discount vouchers for small and medium-sized catering industries; the Japanese government invests a huge amount of funds to subsidize "domestic travel" services provided by the tourism industry, so that tourists can enjoy a 15%~35% discount; In the US, probably the vaccine will be available in the first half of 2021, so that daily traffic is expected to be improved, and the large-scale circulation of consumption vouchers issued by the government will bail the market consumption out of the slump. On the one hand, these policies stimulate consumption and alleviate the business shrinking crisis of SMEs. On the other hand, policies guarantee targeted preferential subsidies for industries that are struck greatly by the pandemic, which saves costs for policy implementation. This can be applied to help SMEs in large cities to come through difficulties and accelerate business development.

Second, focus on reducing administrative barriers, improving the transportation service system, and ensuring the recovery of the business market for SMEs. From the perspective of common practice in market operation, market vitality regulates the operating cycle and operating conditions of enterprises in the industry. The "formalist" market supervision procedures have increased the administrative pressure on enterprises, which has a particularly obvious impact on the survival and development of SMEs. In the early stage of prevention and control of COVID-19, the approval process for resumption was widely used, but its cumbersome and time-consuming procedures increased the "cost" of enterprises to promote the resumption of work, which hindered the resumption of work in SMEs. The current situation gets relatively stable, and the spread effect is relatively controllable. While safeguarding basic public health, the traditional way of governmental supervision in the form of "pre-supervision" tends to trap the market in the dump. Opening the "green channel for resumption of work" for SMEs, and implementing a new resumption filing system of "informed commitment" and "post-supervision" can effectively relax the administrative burden of enterprises and speed up the process of

reorganizing industrial chain and market recovery. At the same time, in response to the problems of public transportation of employees back to work in enterprises, transportation departments should provide support through "point-to-point" customized time and route planning, and strictly follow the pandemic prevention and control arrangements to accurately control the number of passengers, which lays a foundation for the orderly recovery of labor market. The complete social infrastructure and service-oriented policy interventions will guarantee corresponding convenience to SMEs that resume work and production, and reduce the negative impact of pandemic prevention and control, supervision and inspection on market operations.

Third, provide special tax incentives and fiscal policy support to ensure that SMEs can facilitate capital turnover at a lower cost. The government should implement tax reductions and exemptions for SMEs, extend the tax payment period, and provide tax subsidies. In the meantime, expansionary fiscal policies are adopted with liquidity support of direct loans, preferential loans, deferred payment of public utility fees and rents of SMEs. Tax policies can provide general or targeted support for enterprises in the tax system at any time. Fiscal policies provide more effective targeted support for companies that are deeply affected by the crisis and cannot enter the financial system for financing. Combination of the two also helps prevent unnecessary distortions in a single system. Financial expenditure support measures are usually temporary, aimed at helping companies overcome short-term difficulties. As for the long-existing crisis like COVID-19, tax policies are also needed to regulate enterprises with a long-term impact. Among them, for example, UK not only adopts tax policies such as deferred payment of value-added tax and income tax, but also provides small business with subsidies of 10,000 pounds for all enterprises that qualify for small business rate relief or rural rate relief in England; and provides up to 5 million pounds of loans and lower interest rates down to 0.1%. The US provides $377 billion of loans and grants to small businesses with less than 500 employees. To encourage the distribution of these loans, the service fee of bank loans is 5%. Companies can defer payments for one year, and if the money is used to pay employees' salaries, rent, mortgage interest or utility expenses, they can partially waive the loans. These measures shall ensure the survival of sustainable enterprises and guarantee the employment, which are conducive to rapid increase in production and employment when crisis management measures (such as social distancing) are relieved and demand goes up. This will prevent companies from going bankrupt due to lack of liquidity during the crisis. Such bankruptcy is usually very destructive with costly outcomes. At the same time, these policies will also reduce the financial costs of other projects because in the short or medium term, providing support to enterprises will help reduce the unemployment rate and cut down public expenditures on unemployment benefits, social assistance, and wage subsidies.

The negative impact of COVID-19 on the survival and development of SMEs still exists. It is particularly critical to pay attention to the resumption of market circulation and ensure the internal capital liquidity of enterprises. SMEs in megacities in various countries are expected to come through the crisis, with the support of macro policies and regulations.

### 5.1 Limitations and future research

Despite the theoretical and practical implications mentioned above, this study has some limitations. First, although we collected the questionnaires with the help of Federation of Industry and Commerce and Chamber of Commerce in Beijing to try our best to ensure the data quality, there were still improvements in data reliability. All the measures were based on subjective self-assessment, which were affected by individual preferences and perception. Such problem is a common issue in research surveys. In further research, we need to collect some macro data

and explore other methods that can objectively measure "impact degree" to eliminate common method bias and further support the conclusions. Second, this study was conducted under the background of COVID-19 pandemic in Beijing, a representative of megacities. Therefore, the findings may not be generalizable to cities of all sizes and the results are time-limited. Furthermore, the sample size is expected to be enlarged in further research. Last, this study focused on the whole industries, and conclusions were not proposed according to specific industry characteristics. Therefore, future studies can investigate the impact of COVID-19 on small and medium-sized enterprises in a single industry, which will be an in-depth discussion and improvement for this paper.

## Supporting information

**S1 File. Questionnaire in English.**
(DOCX)

**S2 File. Questionnaire in Chinese.**
(DOCX)

## Author Contributions

**Conceptualization:** Zhengwei Ma.

**Data curation:** Yiran Liu.

**Formal analysis:** Yiran Liu.

**Methodology:** Yida Gao.

**Supervision:** Zhengwei Ma.

**Validation:** Yida Gao.

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
