## [Decision Letter · Decision Letter 0]

10 Jun 2021

PONE-D-21-12192

Research on the impact of COVID-19 on Chinese small and medium-sized enterprises: Evidence from Beijing

PLOS ONE

Dear Dr. Zhengwei Ma,

Thank you for submitting your manuscript to PLOS ONE. After careful consideration, we feel that it has merit but does not fully meet PLOS ONE’s publication criteria as it currently stands. Therefore, we invite you to submit a revised version of the manuscript that addresses the points raised during the review process.

We look forward to receiving your revised manuscript.

Kind regards,

László Vasa, PhD

Academic Editor

PLOS ONE

Journal Requirements:

4. Please upload a copy of Figure 1, Figure 2, Figure 3 and Figure 5 to which you refer in your text on page 10, 15, 17, 18, 19 and 20. If the figure is no longer to be included as part of the submission please remove all reference to it within the text.

Additional Editor Comments (if provided):

Reviewers' comments:

Reviewer's Responses to Questions

**Comments to the Author**

1. Is the manuscript technically sound, and do the data support the conclusions?

Reviewer #1: Partly

Reviewer #2: Partly

2. Has the statistical analysis been performed appropriately and rigorously? 

Reviewer #1: No

Reviewer #2: Yes

3. Have the authors made all data underlying the findings in their manuscript fully available?

Reviewer #1: Yes

Reviewer #2: Yes

4. Is the manuscript presented in an intelligible fashion and written in standard English?

Reviewer #1: Yes

Reviewer #2: Yes

5. Review Comments to the Author

Reviewer #1: The topic is actual and calls the attention of the public, but in this form the manuscript is not in compliance with requirements of scientific papers.

The number of hypothesis is too much, it is advised to reconstruct them. The sample (234 valid questionnaires) is too low for drawing conclusions.

The introduction of the results is not visualized, the reviewer did not find the referred figures. The plain text is not transparent enough. Content of Tables would need more clear description.

The research concept is good but presenting the results need major reconstruction.

Reviewer #2: The topic is very much actual - everything that goes about COVID is timely and trendy nowadays. Therefore, investigation on the effects and status of recovery in SME-s in China is a more than welcomed idea and most likely will be of interest to the journal's readers.

The title, abstract, and keywords are acceptable.

In the introduction, however, I have some structural problems: 1.1, the Importance of SME-s is instead belonging to Literature review, so I recommend shifting it there. 1.2 will be enough for introduction, extended by describing the research goals and the context better.

Literature review exists; however, it is unusually detailed articulated, and in some parts, I feel it is instead a chapter of a theoretical book or university booklet. More emphasis should be set on the critical and analytical approach while processing the literature sources.

While there is a methodology chapter, we can see an unusual chapter named the "Hypothesis and construct" chapter, indicating 15 (!) hypotheses. First of all, in my opinion, it should be merged with the methodology chapter, and, on the other hand, 15 hypotheses are just too much. As not all of them belong to the category of accurate and well-defined hypotheses, please merge or cancel a few. In addition, it is just not clear on what basis these hypotheses are defined?

Regarding the results, why these are valid, however, there is a disproportion regarding the structure as it is too short. While the methodology is explained on several pages, the results are on net 2 pages, without a real explanation.

Conclusions include implications and suggestions as well, excellent idea.

There are no limitations indicated in the text; it should be done in the introduction, methodology, or conclusions.

6. PLOS authors have the option to publish the peer review history of their article (what does this mean?). If published, this will include your full peer review and any attached files.

Reviewer #1: No

Reviewer #2: No

---

## [Author Response · Author response to Decision Letter 0]

13 Jul 2021

(1) Respond to reviewer one.

Thank you so much for the reviewers’ comments concerning our manuscript entitled “Research on the Environmental Impact of Tidal Power Generation in China”. Those comments are all valuable and very helpful for revising and improving our manuscript, as well as the important guiding significance to our study. We have studied comments carefully and have made correction which we hope meet with approval. Revised portion are marked in red in the paper. The main corrections in the paper and the responds to the reviewer’s comments are as flowing:

Responds to the reviewer’s comments: 

Reviewer #1: 

1.The number of hypothesis is too much, it is advised to reconstruct them.

Response: Thank you so much for your comment. As there is no well-established SEM that has existed in areas studying the impact of COVID-19 on enterprises, we need to construct the exploratory structural equation model instead of confirmatory structural equation model. Initially we need to consider all possible assumptions that might exist and revise the model after comparison with actual observed data to get the best-fit model[1]. Consequently, the initial hypotheses seem to be too much.

2. The sample (234 valid questionnaires) is too low for drawing conclusions.

Response: Thank you so much for your comment. Bentler[2] has suggested a 5:1 ratio of sample size to number of free parameters to construct a reliable structural equation model and J.Christopher Westland[3] calculated the lower bound of the sample size with 30 indicator variables and 6 latent variables applying algorithm to be 187. Consequently, 234 samples are considered enough for a well-behaved SEM.

3.The introduction of the results is not visualized, the reviewer did not find the referred figures. The plain text is not transparent enough.

Response: Thank you so much for your comment. According to your confusion of the accuracy of the data, we show all the data resources in the Introduction chapter below:

(1)'' Until the end of 2020, the cumulative number of confirmed cases worldwide has reached 91.5 million, and the cumulative death toll has reached 1.96 million. ''

https://international.caixin.com/2021-01-13/101650269.html

(2)'' There has been a reduction in business income of nearly 67.69% of SMEs; 21.61% of SMEs cannot repay loans and other debts in time, facing greater pressure on operating funds; 86.22% of SMEs cannot survive on funds in their accounts for more than 3 months; 33.73% of SMEs do not have enough funds to survive for one month. '' 

https://baijiahao.baidu.com/s?id=1660042030690643188&wfr=spider&for=pc

(3)'' The global economy is expected to shrink by 4.4%, the US GDP will drop by 4.3%, the Eurozone GDP will shrink by 8.3%, and Japan's GDP will drop to 5.3%. '' 

http://www.xhyb.net.cn/news/guoji/2021/0112/141056.html

& https://xueqiu.com/1011660583/160852673

(4)'' The US unemployment rate in October 2020 was 6.9%, with an increase of 3.3% compared to that in October 2019; the EU’s overall unemployment rate rose from 6.6% in September 2019 to 7.5% in September 2020; Japan witnessed a rise of unemployment rate, from 2.4% in September 2019 to 3.0% in September 2020. ''

https://new.qq.com/omn/20201117/20201117A0AFN400.html

& http://ft.newdu.com/economics/word/202101/309995.html

(5)'' The corporate bankruptcy rate in developed countries is expected to increase by 2.4% in 2020 compared to 1.4% in 2019. ''

https://atradius.cn/zh/reports/corporate-insolvency-growth-to-accelerate-in-2020.html

(6)'' 70% of employment opportunities are generated by small and micro enterprises and self-employed individuals. ''

https://www.dx2025.com/archives/85508.html

(7)'' By the end of 2019, Chinese small, micro and medium-sized enterprises accounted for 99.7% of the total number of enterprises in the country. Among them, small and micro enterprises accounted for 97.3%.''

https://wenku.baidu.com/view/1ac0ae3c1b2e453610661ed9ad51f01dc3815754.html

(8)'' Chinese SMEs contribute more than 50% of taxation, more than 60% of GDP, more than 70% of technological innovation, more than 80% of urban labor employment, and more than 90% of the number of enterprises. ''

https://xueshu.baidu.com/usercenter/paper/show?paperid=1b0e00g0k43n0p701j2p0pc0pu131990&site=xueshu_se

(9)'' New York City began the first phase of "unblocking" on June 8th, with more than 400,000 people back to work, and another five areas in New York State will enter the second phase of resumption of work. ''

http://www.21jingji.com/2020/6-9/0NMDEzNzlfMTU2NjU0NA.html

(10)'' The UK started on June 1st, with schools in some areas resuming classes gradually, and "non-essential" retail stores resuming business one after another. ''

https://world.huanqiu.com/gallery/9CaKrnQhZnj

(11)'' In Japan, since August 1st, basic restrictions have been completely lifted, and work has resumed. ''

http://cifer.pbcsf.tsinghua.edu.cn/index.php?m=content&c=index&a=show&catid=113&id=473

(12)'' As of July 2nd, various industries in China have basically resumed production, with a resumption rate of 99.1%, and on average, 95.4% of people have resumed their work. More than 50 cities have resumed full operations. Among them, the resumption rate of SMEs has reached 91%, which has fully promoted the recovery of production capacity and directly affected economic growth. ''

https://app.21jingji.com/html/2020yiqing_fgfc/

(13)'' In 2020, China's GDP is expected to grow at 2.3%, ranking first in the world. '' 

http://cn.dailyeconomic.com/finance/2021/01/18/22370.html

4.Content of Tables would need more clear description.

Response: Thank you so much for your comment. We add the ''Nomenclature'' before chapter ''Introduction''. Please review it. Revised portions are marked in red in the paper.

Thank you so much for your comments. I really learn a lot for your comments. All comments are very valuable and helpful for revising and improving our manuscript, as well as the important guiding significance to our study.

References:

[1]Jöreskog, K.G. & Sörbom, D. (1996). LISREL 8 User's reference guide. Chicago: Scientific Software.

[2]Bentler, P. M. EQS, Structural Equations, Program Manual, Program Version 3.0, BMDP Statistical Software, Inc., Los Angeles, 1989, 6.

[3]J. Christopher Westland. Lower bounds on sample size in structural equation modeling[J]. Electronic Commerce Research and Applications,2010,9(6).

(2) respond to reviewer two.

Thank you so much for the reviewers’ comments concerning our manuscript entitled “Research on the Environmental Impact of Tidal Power Generation in China”. Those comments are all valuable and very helpful for revising and improving our manuscript, as well as the important guiding significance to our study. We have studied comments carefully and have made correction which we hope meet with approval. Revised portion are marked in red in the paper. The main corrections in the paper and the responds to the reviewer’s comments are as flowing:

Responds to the reviewer’s comments: 

Reviewer #2: 

1. 1.1, the Importance of SME-s is instead belonging to Literature review, so I recommend shifting it there. 1.2 will be enough for introduction, extended by describing the research goals and the context better.

Response: Thank you so much for your comment. We have moved 1.1 to Literature review and revised the subtitle of 1.2. Revised portions are marked in red in the paper.

2.Literature review exists; however, it is unusually detailed articulated, and in some parts, I feel it is instead a chapter of a theoretical book or university booklet. More emphasis should be set on the critical and analytical approach while processing the literature sources.

Response: Thank you so much for your comment. Following your suggestion, we have revised the literature review. We add the new chapter "Analytical approach" in literature review. Please review it. Revised portions are marked in red in the paper.

3.While there is a methodology chapter, we can see an unusual chapter named the "Hypothesis and construct" chapter, indicating 15 (!) hypotheses. First of all, in my opinion, it should be merged with the methodology chapter.

Response: Thank you so much for your comment. Following your suggestion, we have merged "Hypothesis and analysis" chapter with "methodology" chapter. Please review it. Revised portions are marked in red in the paper.

4.On the other hand, 15 hypotheses are just too much. As not all of them belong to the category of accurate and well-defined hypotheses, please merge or cancel a few. 

Response: Thank you so much for your comment. As there is no well-established SEM that has existed in areas studying the impact of COVID-19 on enterprises, we need to construct the exploratory structural equation model instead of confirmatory structural equation model. Initially we need to consider all possible assumptions that might exist and revise the model after comparison with actual observed data to get the best-fit model[1]. Consequently, the initial hypotheses seem to be too much.

5.In addition, it is just not clear on what basis these hypotheses are defined.

Response: Thank you so much for your comment. We reconstruct chapter 3.1.2 "Hypothesis" and add the theoretical basis for these hypothetical definitions. Please review it.

6.Regarding the results, why these are valid, however, there is a disproportion regarding the structure as it is too short. While the methodology is explained on several pages, the results are on net 2 pages, without a real explanation.

Response: Thank you so much for your comment. We have rewritten the chapter “Results”. Revised portion are marked in red in the paper. Please review it.

7.There are no limitations indicated in the text; it should be done in the introduction, methodology, or conclusions.

Response: Thank you so much for your comment. We add the new chapter "Limitations and future research" before chapter "Conflicts of Interest". Please review it.

Thank you so much for your comments. I really learn a lot for your comments. All comments are very valuable and helpful for revising and improving our manuscript, as well as the important guiding significance to our study.

References:

[1]Jöreskog, K.G. & Sörbom, D. (1996). LISREL 8 User's reference guide. Chicago: Scientific Software.

---

## [Decision Letter · Decision Letter 1]

28 Jul 2021

PONE-D-21-12192R1

Research on the impact of COVID-19 on Chinese small and medium-sized enterprises: Evidence from Beijing

PLOS ONE

Dear Dr. Zhengwei Ma,

Thank you for submitting your manuscript to PLOS ONE. After careful consideration, we feel that it has merit but does not fully meet PLOS ONE’s publication criteria as it currently stands. Therefore, we invite you to submit a revised version of the manuscript that addresses the points raised during the review process.

We look forward to receiving your revised manuscript.

Kind regards,

László Vasa, PhD

Academic Editor

PLOS ONE

Journal Requirements:

Reviewers' comments:

Reviewer's Responses to Questions

**Comments to the Author**

1. If the authors have adequately addressed your comments raised in a previous round of review and you feel that this manuscript is now acceptable for publication, you may indicate that here to bypass the “Comments to the Author” section, enter your conflict of interest statement in the “Confidential to Editor” section, and submit your "Accept" recommendation.

Reviewer #1: All comments have been addressed

Reviewer #2: (No Response)

2. Is the manuscript technically sound, and do the data support the conclusions?

Reviewer #1: Yes

Reviewer #2: Partly

3. Has the statistical analysis been performed appropriately and rigorously? 

Reviewer #1: Yes

Reviewer #2: Yes

4. Have the authors made all data underlying the findings in their manuscript fully available?

Reviewer #1: Yes

Reviewer #2: Yes

5. Is the manuscript presented in an intelligible fashion and written in standard English?

Reviewer #1: Yes

Reviewer #2: Yes

6. Review Comments to the Author

Reviewer #1: The authors answered the comments of the reviewers, the structure became more clear. I am satisfied by the corrections and in this form the manuscript is recommended for publication.

Reviewer #2: The authors improved the paper based on the reviewer's suggestions; in some cases just explained and argued for their original solutions. The papaer is much better in its current version, however, I still have concerns regarding the large number of hypotheses, I stil revommend to lower the number of hypotheses, deleting and merging some.

This review was made by the academic editor.

7. PLOS authors have the option to publish the peer review history of their article (what does this mean?). If published, this will include your full peer review and any attached files.

Reviewer #1: No

Reviewer #2: No

---

## [Author Response · Author response to Decision Letter 1]

2 Aug 2021

Thank you so much for the your comment concerning our manuscript entitled “Research on the impact of COVID-19 on Chinese small and medium-sized enterprises : Evidence from Beijing”. The comment is valuable and very helpful for revising and improving our manuscript, as well as the important guiding significance to our study. We have studied the comment carefully and have made correction which we hope meet with approval. Revised portion are marked in red in the paper. The main corrections in the paper and the respond to your comment is as flowing:

Responds to the reviewer’s comment: 

 Reviewer #2: 

 1.The paper is much better in its current version, however, I still have concerns regarding the large number of hypotheses, I still recommend to lower the number of hypotheses, deleting and merging some. 

 Response: Thank you so much for your comment. We have deleted and merged some hypotheses and restructured the model. Revised portions are marked in red in the paper.

 Thank you again for your comment. I really learn a lot for your comment. The comment is very valuable and helpful for revising and improving our manuscript, as well as the important guiding significance to our study.

---

## [Decision Letter · Decision Letter 2]

23 Aug 2021

Research on the impact of COVID-19 on Chinese small and medium-sized enterprises: Evidence from Beijing

PONE-D-21-12192R2

Dear Dr. Zhengwei Ma,

We’re pleased to inform you that your manuscript has been judged scientifically suitable for publication and will be formally accepted for publication once it meets all outstanding technical requirements.

Kind regards,

László Vasa, PhD

Academic Editor

PLOS ONE

Additional Editor Comments (optional):

Reviewers' comments:

Reviewer's Responses to Questions

**Comments to the Author**

1. If the authors have adequately addressed your comments raised in a previous round of review and you feel that this manuscript is now acceptable for publication, you may indicate that here to bypass the “Comments to the Author” section, enter your conflict of interest statement in the “Confidential to Editor” section, and submit your "Accept" recommendation.

Reviewer #1: All comments have been addressed

Reviewer #2: All comments have been addressed

2. Is the manuscript technically sound, and do the data support the conclusions?

Reviewer #1: Yes

Reviewer #2: Yes

3. Has the statistical analysis been performed appropriately and rigorously? 

Reviewer #1: Yes

Reviewer #2: Yes

4. Have the authors made all data underlying the findings in their manuscript fully available?

Reviewer #1: Yes

Reviewer #2: Yes

5. Is the manuscript presented in an intelligible fashion and written in standard English?

Reviewer #1: Yes

Reviewer #2: Yes

6. Review Comments to the Author

Reviewer #1: Some remarks for finalization: the words "hypothesis" - singular - and "hypotheses" - plural - should be checked, it is wrongly written in some parts of the manuscript (for example, 3., 3.1, 3.1.2 chapter titles). Due to the reconstruction of hypotheses, now H6 is right after H3. It is understandable for the reviewers, who followed the review process, but readers can hardly understand this structure. I recommend to check it and correct if possible.

Reviewer #2: The authors improved their paper taking my recommendations into consideration. The paper is much better and sound in its current form. As all suggestions were considered and implemented, I recommend the paper for publication in its current form.

7. PLOS authors have the option to publish the peer review history of their article (what does this mean?). If published, this will include your full peer review and any attached files.

Reviewer #1: No

Reviewer #2: No

---

## [Editor Report · Acceptance letter]

9 Nov 2021

PONE-D-21-12192R2 

Research on the impact of COVID-19 on Chinese small and medium-sized enterprises: Evidence from Beijing 

Dear Dr. Ma:

I'm pleased to inform you that your manuscript has been deemed suitable for publication in PLOS ONE. Congratulations! Your manuscript is now with our production department. 

Kind regards, 

on behalf of

Prof. Dr. László Vasa 

Academic Editor

PLOS ONE